# Refining animal care through technology: Addressing alopecia in *Jaculus jaculus* with validated computer vision analysis

Matthew D. Boulanger[1,2], Juri A. Miyamae[2], Tara Martin[1,3], Gerry Hish[1], Talia Y. Moore[2,4]*

1 Unit for Laboratory Animal Medicine, University of Michigan, Ann Arbor, Michigan, United States of America, 2 Department of Robotics, University of Michigan, Ann Arbor, Michigan, United States of America, 3 Refinement & Enrichment Advancements Laboratory, University of Michigan, Ann Arbor, Michigan, United States of America, 4 Department of Mechanical Engineering, Ecology and Evolutionary Biology, Museum of Zoology, University of Michigan, Ann Arbor, Michigan, United States of America

* taliaym@umich.edu

## Abstract

We validated the use of an open-source computer vision toolkit to analyze high-quality behavioral data and evaluate welfare in the Lesser Egyptian Jerboa (*Jaculus jaculus*). Movements of these small, nocturnal rodents are rapid and difficult to observe, potentially obscuring behavioral assessment. However, assessment became warranted when alopecia and jumping were noted. We compared trained human observers to machine learning trained computer vision algorithms, evaluating accuracy and precision in behavioral classification. Human observers categorized behaviors with an overall accuracy of $0.71 \pm 0.11$ and an intraclass correlation coefficient (ICC) of $0.92 \pm 0.07$, with greater odds of misidentifying behaviors lasting less than one second. Computer vision classifiers successfully met human-grade accuracy and ICC, with significantly less sensitivity to behavioral duration. As 34% of manually classified behaviors lasted less than 0.5 seconds, we used computer vision to annotate activity budgets of captive jerboas before and after adding novel enrichment. Alopecia was significantly associated with grooming, and while grooming was negatively associated with terrarium height and with opaque dividers between terraria, conventional rodent enrichment had no significant effect on behavior. Inflammatory causes of alopecia were not found with cytologic, molecular, or histopathologic analysis. These results suggest captive jerboa may demonstrate psychogenic alopecia. Furthermore, computer vision automation allows for fast, accurate analysis of large volumes of behavioral data that can be used to tailor species-specific husbandry practices and improve animal welfare.

**Data availability statement:** All relevant data supporting the findings of this study are available within the article and its supplementary information files. The dataset used in this study is available in the University of Michigan's Deep Blue Data repository, accessible at https://doi.org/10.7302/3mqd-8743. Currently this dataset is under embargo, but will become publicly accessible if accepted and can be made available to reviewers.

**Funding:** The author(s) received no specific funding for this work.

**Competing interests:** The authors have declared that no competing interests exist.

## Introduction

Jerboas (family: Dipodidae) are desert rodents and obligate bipeds with extreme morphological and ecological adaptations [1,2]. Jerboas engage in rapid, unpredictable ricochetal locomotion to evade predators, such as vertical leaping to avoid snakes [3,4]. Previous work has demonstrated that increased territorial range was associated with hindlimb elongation, that the most proficient leapers inhabit the most sparsely vegetated deserts, and that jerboa species travel significantly farther than other rodents prior to weaning [5,6]. These major locomotory differences from mice and rats, despite their relatively close phylogenetic relationship, make jerboas a compelling model to study the evolutionary processes, developmental mechanisms, and biomechanical as well as ecological consequences of limb specialization. However, this ecological context should be considered when working with the species in captivity.

In our research colony of Lesser Egyptian Jerboa (*Jaculus jaculus*), we noted two behaviors that merited further investigation: jumping and alopecia without documented clinical cause. In other rodent species, such as the gerbil (*Meriones unguiculatus*), repetitive corner jumping can be a stereotypic behavior [7]. Abnormal repetitive behaviors, including stereotypies and compulsions, can indicate a negative welfare state [8–10]. In these cases, animals compensate for distress by expressing behaviors that otherwise have rewarding associations in a maladaptive manner. Examples in other rodent species include corner digging, jumping, and psychogenic alopecia [7,11]. It is unknown whether captive jerboas engage in jumping as an abnormal repetitive behavior, or whether this is related to maintaining physical condition, as observed in the wild [3]. Jerboa leaps are predominantly powered by muscle contractions, rather than benefiting from passive energy recycling by tendons [12]. Muscles are dynamic and can atrophy without use [13], so jerboas may engage in "exercise-like" behaviors to maintain predator evasion ability. Wild jerboa pups have been observed hopping outside their burrow, which is suggested as a "calisthenic event" to prepare for the desert wilds [14]. Adult jerboas may recapitulate this activity in captivity to maintain physical condition. Rather than assuming that all repeated behaviors indicate a negative welfare state, this ecological context motivates explicit study.

Effective environmental enrichment may promote a sense of environmental control (autonomy) and allow for expression of innate behaviors [8]. Enrichment items should be safe, goal-oriented, and utilized by the animal; emphasizing the need for knowledge of the animal's natural history and resource perspective [2,13–15]. We set out to determine whether jumping and grooming are indicators of distress in jerboas, and if so, to assess whether environmental enrichment would result in mitigation.

However, meeting these ideal parameters is not straightforward, as direct inquiry of animal subjects is currently impossible. Researchers and veterinarians interested in assessing behavior must instead often rely on interpreting large quantities of observational data. Due to their adaptations for rapid movement, jerboas are difficult to observe cageside. Computer vision and machine learning techniques have enhanced the efficiency and objectivity of animal behavior studies by training neural networks to correlate visual features with behavioral categories [16–19]. Automated analysis enables researchers to leverage larger datasets, yielding more holistic and

macroscopic examinations of behavior. This saves researcher time by eliminating human processing and interpretation, and ideally yields more accurate and repeatable results [20].

Here, we used computer vision to automatically classify the behaviors of captive jerboas recorded remotely with a Raspberry Pi camera system. We analyzed those data to understand how different aspects of husbandry modulate the frequency and distribution of their behaviors. We used DeepAction, an open-source Matlab toolbox, to enable automated classification of behaviors from videos. This package uses convolutional neural networks (CNNs) and dense optic flow to extract both spatial (from raw pixel values) and temporal (from comparing adjacent frames) data for each frame of video to train a long short-term memory (LSTM) classifier to predict behavior [17].

This study was conducted in three phases. First, trained human observation of jerboa behaviors exhibited within standard husbandry were used to test hypothesis $H_1$: given their inherently repetitive nature, grooming and jumping occupy the largest portion of captive jerboa's activity budget; and $H_2$: alopecia is associated with grooming behaviors. Second, we used the standard husbandry dataset to evaluate automated tools by testing $H_3$: trained machine classifiers can be as accurate and precise as human observers scoring jerboa behavior. Third, with benchmarks for classification performance, we introduced novel enrichment items to jerboas and measured their influence on classified activity budgets to test hypothesis $H_4$: environmental manipulations, including visual separation and provision of manipulanda, will decrease the frequency of repetitive behaviors.

## Methods

### Animal husbandry and recording techniques

All animals were housed in an AAALAC International-accredited facility in accordance with the Animal Welfare Act and the *Guide for the Care and Use of Laboratory Animals* [21,22]. All work was performed with the approval of the University of Michigan's Institutional Animal Care and Use Committee (IACUC).

Lesser Egyptian Jerboas were obtained two years prior to the study start date from another AAALAC-accredited institution to establish a breeding colony. Husbandry follows previously established protocols [23], including providing a dust bath, complex diet, nesting materials (Nestlets, Ancare), and Timothy hay, which are considered standardized sources of enrichment for this species. Animals had olfactory, visual, and auditory contact with conspecifics. During the study period, the colony consisted of approximately 20 individuals that were housed individually, as breeding pairs, or as dams with pups in a climate-controlled room with a 16:8-h light:dark cycle (lights on at 22:30 and off at 14:30). Animals used in this study were non-breeding adults, singly housed in either 15 or 20 gallon glass terrariums: either ~35 cm or ~51 cm tall, respectively, with identical floor space (~1900 cm$^2$) (Aqueon Products, Franklin WI).

Infrared (IR) light sources were used to illuminate the dark period. We recorded singly housed adult jerboa behaviors using cameras without an IR filter (Camera Module 3 and Raspberry Pi 4, Raspberry Pi foundation, Cambridge, England), set with a fixed focal length at 30 fps and 1296 x 1600 resolution. All recordings were initiated remotely, with a minimum of 30 minutes following previous room entries and without a human in the room to avoid effects of observer presence on behavior [24]. A single human rater (M.D.B.) performed all initial manual annotations of jerboa behavior.

To the authors' knowledge, jerboa behavior has not been comprehensively characterized in the wild or in captivity. A novel baseline ethogram was established. For exhaustive ethograms, wherein every unique behavior was described, two adult (one male and female) singly housed jerboas were recorded on two different days each for 24 hours and reviewed by the rater. Previous work suggests that wild jerboas are nocturnal [4,14,23], therefore, all subsequent recordings occurred during the dark phase, generally between one to two hours after the onset. For experimental observations, we developed a mutually exclusive ethogram (S1 Table) that encompassed the most common exhaustive behaviors exhibited by animals, but still accounted for all behaviors observed. For example, uncommon behaviors such as foot drumming [3] were incorporated into the 'upright, rapid movements' state.

## Behavioral surveillance

To determine the average inter-individual distribution of behaviors, eleven adults were then recorded for a single 30 minute session (one to two hours after the onset of the dark period.) All videos were manually annotated by the rater. The influence of multiple parameters on behavior (animal age, sex, rack position, terrarium height) prior to experimental intervention was also considered.

## Alopecia surveillance

At the time of recording, about one third of the colony was being monitored for clinical alopecia without an identified etiology. Therefore, multiple potential etiopathogeneses were screened for.

To assess potential parasitic causes of alopecia, we collected fecal samples and swabbed the substrate interface, along dust baths, and nest boxes of 13 terrariums containing 15 animals. Additionally, two animals with clinical alopecia were directly swabbed and their feces was pooled for the same tests. These samples were submitted for Demodex, mite, and pinworm PCR infectious agent panels (PRIA, Charles River Laboratories, Wilmington, MA).

We developed a jerboa alopecia scale based on previously established non-human primate methods [25]. Alopecia is scored from 0-3, with 0 representing a fully intact coat, 3 representing totally bare skin, and 1–2 representing intermediate scores. To better understand distribution, we divided the body into zones: the dorsal region between the shoulders and back, the tail base including the tail, and both hindlimbs considered together as a single zone (Fig 1). In other rodent species, normal grooming progresses cephalocaudally [26], but repetitive grooming of particular areas can occur in times of stress, in both singly and group housed animals, which can lead to alopecia ("barbering") [27,28]. For semi-quantitative assessment of the effect of behavior, a binary presence/absence scoring of alopecia in any zone was considered.

A single animal with grade 3 alopecia was euthanized for an unrelated medical condition (Fig 1) and was submitted for gross necropsy and microscopic examination. Tissue samples were fixed for 24 hours in 10% neutral buffered formalin. These sections were embedded in paraffin wax, stained with hematoxylin and eosin, and evaluated with routine light microscopy by a board certified veterinary pathologist. At the time of necropsy, fur plucks as well as deep and superficial skin scrapes were prepared for routine cytologic light microscopic inspection.

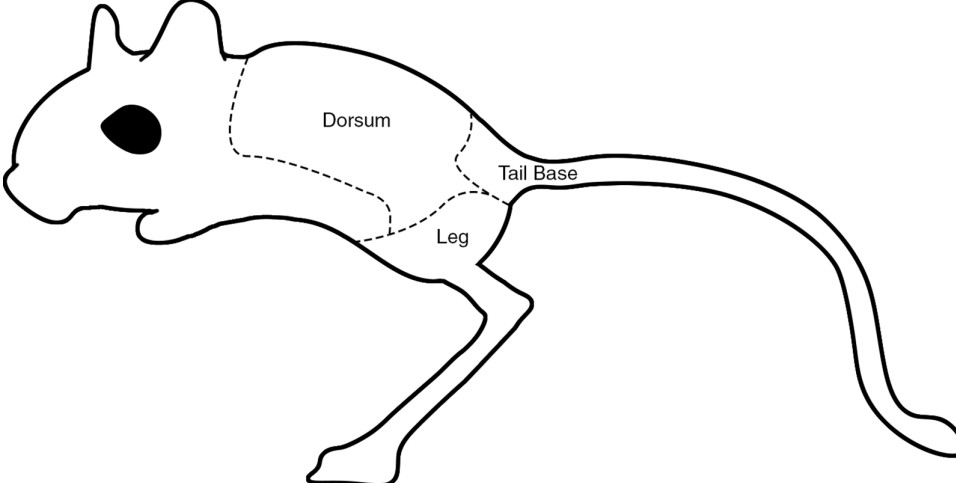

**Fig 1. Scoring and quantification of jerboa alopecia.** Alopecia was graded by anatomic location and character. Anatomic locations include dorsum (the area between the shoulders and the back), tail base (tail base and tail), and legs (as a single zone). Character score ordinally ranges from 0 (no alopecia noted) to 3 (completely bare skin.).

To determine behavioral correlates with alopecia, we examined 12 five-minute clips of eight individual jerboas with standard husbandry conditions and graded alopecia. We then evaluated a binomial logistic model that predicts the presence or absence of alopecia (binary state) as a function of the type and duration of behaviors while considering animal identity as a random effect.

## Quantification of human observer performance and validation of automated machine classifiers

We evaluated human accuracy on a subsample of the activity budget dataset to determine a threshold accuracy for the automated ethogram classifier. From the manually annotated activity budget dataset, we sampled a comparison subset (Fig 2, 'CS') that included 12 one-minute clips: eight unique one-minute clips from five individual jerboas (selected for equal representation of behaviors) and a subset of four horizontally-mirrored clips to assess precision. The same comparison subset was used to evaluate human observations, classifiers trained on the activity budget dataset, and the inference performed by the fully trained automated classifier.

From the activity budget dataset (Fig 2A), we sampled a training subset that included 360 minutes of annotated video, encompassing ten adult singly housed jerboas under standard husbandry (Fig 2A). The neural network was trained with the training subset for up to a maximum of 16 epochs, wherein clips were distributed into 95% for training, 2.5% testing, and 2.5% validation within each epoch (Fig 2A). Within each epoch, a mini-batch of eight was utilized, for a maximum of 45 steps, and the initial learning rate was set to 0.001. Validation occurred once per epoch, and to reduce overfitting the patience for validation loss was set to 2 with a learning rate drop factor of 0.1 every 4 epochs. The training process was repeated ten additional times with new, previously untrained classifiers to generate results for eleven independent

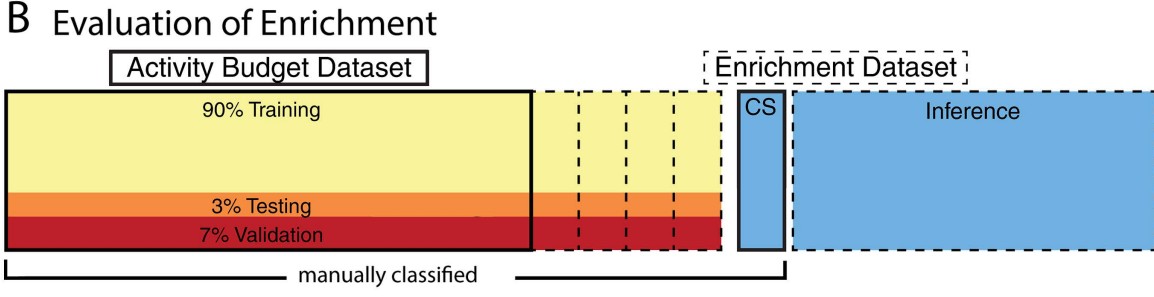

**Fig 2. Full use of video data.** a) The accuracy of human observers was determined to create a performance-based benchmark for classifiers. Data utilized to quantify the activity budget of jerboa with standard husbandry (solid line) was utilized as the training subset for classifier validation and examples of each behavior were sampled for human orientation (hash). b) Classifiers were then fully trained to make inferences on the larger enrichment dataset. Twelve minutes of clips were utilized as a comparison subset between human observers and machine classifiers (blue, 'CS'). To evaluate enrichment, classifiers were trained with confidence based review in additional half hour increments (dashed lines) to make the necessary enrichment subsets for full classifier training. These were then held to human-grade performance standards, derived from the validation phase, to designate sufficient predictions and tested against the same comparison set.

machines. This process did not involve post-hoc confidence-based reviews or repeat training of an individual machine classifier. Each classifier then made inferences of the same comparison subset.

To gauge accuracy and precision of human observers scoring jerboa behaviors, eleven human test observers evaluated video clips after a standardized orientation protocol. Each observer was trained with the same pre-recorded, self-paced presentation and allowed as much time as needed to review the material. From the training subset, we sampled an orientation subset that included video examples of every behavior sourced from activity budget data (six individual jerboas under standard husbandry (Fig 2A, cross-hatched regions)). The orientation included the orientation subset along with explicit, written definitions for all mutually exclusive behaviors. This orientation presentation remained available to observers for reference while scoring.

To evaluate performance of human observers, we asked them to encode the comparison subset (Fig 2, 'CS'). Observers were blinded to the identity of each individual jerboa, and comparison subset videos were presented in random order to each different observer. Observers used the graphical interface for manual scoring included in the DeepAction software package to annotate videos. Observer time usage was not restricted and they could slow down or speed up videos at will with the DeepAction interface.

Accuracy of observer scores was evaluated by categorizing eight unique clips in the testing set and calculated as percentage of frames wherein the observer's annotation was identical to a gold standard rater (Eq. 1). Precision of observer scores was calculated by comparing four of the unique clips with their horizontal mirror to determine intraclass correlation (ICC) for each video frame [29]. This effectively determined intrarater variability by assessing how consistently observers assigned behavior scores to the same video content, albeit presented in a different directional orientation.

$$Accuracy = \frac{\sum_{start\ frame}^{end\ frame} behavioral\ state_{frame,observer} = behavioral\ state_{frame,rater}}{total\ number\ of\ frames} \tag{1}$$

The duration of individual behaviors in each testing clip was calculated by the number of consecutive frames, and the mean duration of each behavior was determined within each clip. Accuracy of either human observers or fully trained machine classifiers was then determined for each behavior within each testing clip.

## Evaluation of Enrichment

After the activity budget data under standard husbandry were collected as baseline, eight animals were exposed to up to three different items for a maximum 14 days per item in a randomized order, with a minimum of one week washout period between items (S2 Table). Items were offered one at a time and included: a stick of manzanita wood (Bio-Serv, Flemington, NJ), a cardboard tunnel (Bio-tunnels for mice, Bio-Serv, Flemington, NJ), and an opaque divider placed between terrariums (Fig 3). Two new behavioral states (interaction with stick, interaction with tunnel) were added to the previous exclusive ethogram to reflect novel item interactions (S3 Table). These items were implemented to test either standard rodent enrichment items (stick, tunnel), or social management by limiting adjacent conspecific visual access (divider), as has been implemented in other solitary species [30].

To incorporate new behaviors associated with enrichment items, we added annotated enrichment videos to the training subset to create new enrichment subsets. The enrichment subsets were then used to train DeepAction-based classifiers using supervised learning and confidence-based review [17]. Enrichment subsets were proportioned into 90%, 7%, and 3% as testing, validation, and training, respectively, for up to 16 epochs (Fig 2B). Within each epoch a mini-batch of 8 was utilized, for a maximum of 45 steps, and the initial learning rate was set to 0.001. Validation occurred once per epoch; patience for validation loss was set to 2 with a learning rate drop factor of 0.1 every 4 epochs to reduce overfitting. If the overall neural network accuracy was below 0.71, we evaluated the accuracy of each unique behavior. We then selected 30 minutes of video clips that included the lowest accuracy behaviors to add to the enrichment subset (Fig 2B), then

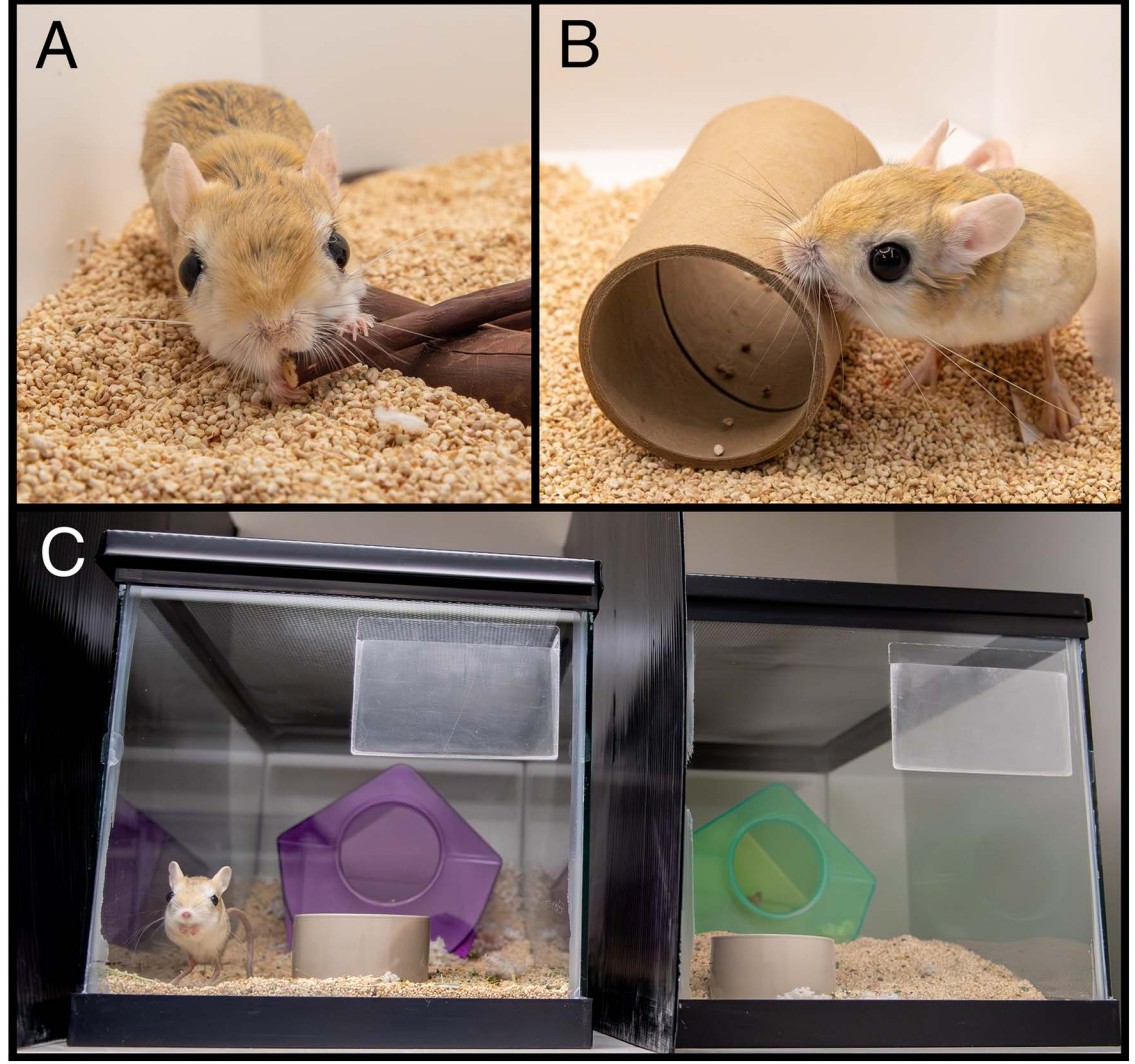

**Fig 3. Novel environmental manipulations.** Singly housed adult jerboas were offered either a) a manzanita wood stick, b) a cardboard tunnel, c) or an opaque divider placed between housing units.

began a new training bout. We repeated this process until the overall accuracy was above 0.71. The final classifier was trained on 460 minutes of videos, achieving an accuracy of 0.73 based on the DeepAction's metrics as seen in Table 1 [17]. We constructed generalized linear mixed models (GLMM) that used fixed effects (behavioral state and presence of enrichment items) and a random effect (animal identity) to predict the duration of the behavior, measured in number of frames.

## Statistical reporting

All statistical analyses were performed with Matlab Statistics and Machine Learning Toolbox (Version 21.4, Mathworks Inc., Natick, MA). Significance was set to $p = 0.05$ and results will be displayed as mean ± standard deviation. For models, 95% confidence intervals (CI) and estimates for fixed effects are reported alongside probability values. Normality was determined with gross inspection of data distribution and histograms. Independent or paired, as appropriate, Student's T-tests

**Table 1. The fully trained computer vision classifier was 0.78 accurate overall, according to internal metrics [17], but accuracy varied among behaviors. Notably, machine classification of jumping and grooming was more accurate than human observers (Fig 5).**

| Behavioral State | Accuracy | True Positive Rate | False Positive Rate |
|---|---|---|---|
| Ambulating | 0.61 | 0.65 | 0.02 |
| Upright, Rapid Movements | 0.85 | 0.88 | 0.07 |
| Food Bowl Interactions | 0.98 | 0.96 | 0.002 |
| Wall Interactions | 0.88 | 0.83 | 0.01 |
| Jumping | 0.89 | 0.88 | 0.006 |
| Dust Bath Interactions | 0.95 | 0.97 | 0.006 |
| Grooming | 0.75 | 0.70 | 0.025 |
| Rolling | 0.99 | 0.67 | 0.00004 |
| Obscured | 0.99 | 0.99 | 0.002 |
| Tunnel Interactions | 0.77 | 0.72 | 0.006 |
| Wooden Item Interactions | 0.68 | 0.6 | 0.003 |

were performed assuming equal variances with two-tailed data for the majority of statistical assessment. To determine the odds ratio of accuracy as a function of behavioral duration, data were made categorical with thresholding and Fisher's Exact Test was used to assess the associated contingency table. Individual jerboa information is detailed in S4 Table.

## Results

### Jerboas most frequently participated in feed-seeking behaviors

Animals spent most of their time ($34.8 \pm 15.8\%$) participating in rapid, upright movements (RM) (Fig 4). This included behaviors where the animal was upright and either non-ambulatory or single stepping, such as foraging (S1 Table). Similarly, the second-most ($11.4 + 6.8\%$) frequently demonstrated behavior was food bowl interactions (FB), which involves direct feed-seeking. Wall interactions, grooming, and jumping made up $10.5 \pm 10\%$, $10 \pm 8.9\%$, and $7 \pm 10.4\%$ of animals' time, respectively. These results refute $H_1$ by revealing that grooming and jumping do not occupy the majority of the activity budget, even when summed together. Instead, the third-most ($11.3 + 14.9\%$) demonstrated behavior was ambulating (AM), suggesting jerboas prioritize exploring their environment. Animals in ~51 cm tall (n = 4) terrariums groomed significantly less than those in ~35 cm (n = 7) terrariums (p = 0.01, $\beta$ = 0.19, independent t-test) occupying $2.14 \pm 2.95\%$ and $14.22 \pm 8.52\%$ of the observed period, respectively. Across all behaviors, standard deviation of incidence was $11.26 \pm 4.3\%$.

### Alopecia is associated with behavioral grooming

Alopecia was noted in eight animals at baseline recording. Four animals demonstrated low grade (1) alopecia in a single zone, either along the dorsum or tail base, with each representing 50% incidence. High grade (3) alopecia was only noted in one animal, at the tail base, emphasizing this location as a site to monitor for significant progression of alopecia in jerboas. Because alopecia is often progressive in other species [25,31], consistent with our clinical experience in jerboas, this suggests lesions may begin either at the tail base or along the dorsum. Dorsal alopecia, regardless of grade, accounted for 47.3% of all alopecia scores, whereas the tail base and leg regions demonstrated 36.8% and 15.8% incidences, respectively.

Samples collected from the entire colony were negative for screening of: *Demodex aurati*, mites (*Myobia musculi, M. musculinus, Radfordia affinis, R. ensifera*), and pinworms (*Aspiculuris tetraptera, Syphacia muris, S. obvelata*). No inflammatory or infectious agents were identified via cytology, gross necropsy, or molecular evaluation of the skin. Alopecic regions demonstrated moderate hair follicle atrophy and epidermal hypoplasia on histopathology. Clinically, animals did not demonstrate signs of pruritus, such as scratching, biting, or linear lacerations.

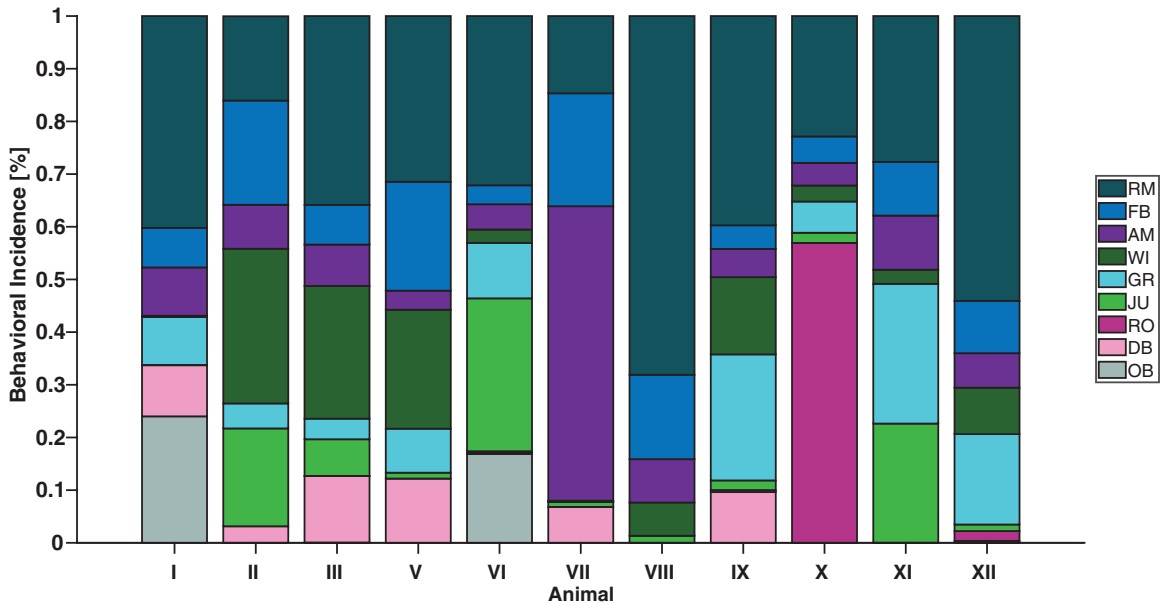

**Fig 4. Adult jerboas spend the majority of their time with food-directed behaviors.** Both the first and second most frequently demonstrated behaviors involve food-seeking, including RM and FB (34.8 ± 15.8% and 11.4 + 6.8%, respectively). Activity budgets of multiple jerboas observed for 30 minutes during the most active period, n = 11. Descriptions can be found in S1 Table. (RM = "Rapid, Upright Movements," FB = "Food Bowl Interactions," AM = "Ambulating," WI = "Wall Interactions," GR = "Grooming," JU = "Jumping," RO = "Rolling," DB = "Dust Bath Interactions," OB = "Obstructed").

Given lack of evidence for alternative causes for alopecia, we evaluated behavioral grooming. A binomial logistic regression ($R^2$ = 0.99) reported a significant relationship between behavioral grooming and the presence of alopecia, both in terms of the amount of grooming (p = $2 \times 10^{-9}$, CI [1.2 2.5], estimate = 1.8) and presence of grooming (p = $5 \times 10^{-6}$, CI [−3.7 −1.5], estimate = −2.6), with individual identity as a random factor. In other words, animals that groomed more frequently were more likely to have alopecia. To further clarify the influence of grooming in isolation from other behaviors, we evaluated a subset of the dataset that only included grooming data, alopecia, and individual identity. This model indicated a similar association of alopecia with the amount of grooming ($R^2$ = 0.96, p = $3 \times 10^{-8}$, CI [1.1 2.25], estimate = 1.6). Altogether, these results support $H_2$: alopecia is associated with grooming behaviors.

Ethogram behaviors are mutually exclusive, meaning increasing the duration of one behavior leads to decreases in other behaviors. To consider the effect of jumping, we examined an equivalent subset of the data that only included jumping, alopecia, and individual identity. This model demonstrated less goodness-of-fit and suggested that jumping is negatively associated with alopecia ($R^2$ = 0.92, p = 0.000032, CI [−1.12 −0.41], estimate = −0.76). This trend suggests that animals without alopecia may spend more time performing other behaviors, including jumping.

### Computer vision reaches human-grade behavioral identification

On average, humans observing the comparison dataset achieved an overall accuracy of 0.71 ± 0.11 and precision (intra-class correlation coefficient) of 0.93 ± 0.069 across all behaviors (Fig 5, orange). Machine classifiers achieved an overall accuracy of 0.76 ± 0.054, with precision of 0.89 ± 0.069, when scoring the same testing set (Fig 5, blue). The distributions of accuracy rates for both human observers and machine classifiers were bimodal, with the lowest scoring machine classifiers performing better than the lowest scoring humans (Fig 5B). No statistically significant difference was found between average accuracy or precision of data generated by observers versus classifiers (*p* >> 0.05). These results support $H_3$: machine classifiers, such as DeepAction, reach the accuracy and precision of human observers at scoring captive jerboa behavior.

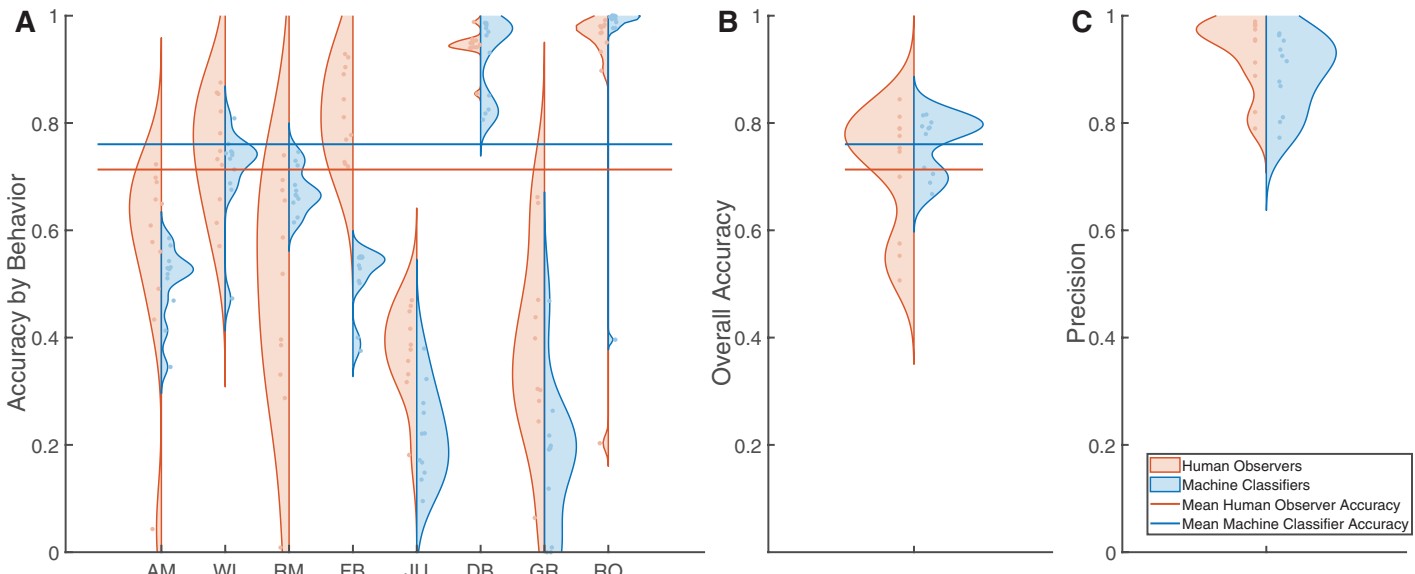

**Fig 5. Machine classifiers successfully met human-grade performance standards, including both accuracy and precision.** a) After orientation, human observers (n = 11, orange) annotated jerboa behavior in the comparison subset. Adequately trained independent machine classifiers (n = 11, blue) also annotated the same comparison subset (AM = Ambulating, WI = Wall Interaction, RM = Rapid, Upright Movements, FB = Food Bowl Interactions, JU = Jumping, DB = Dust Bath Interactions, GR = Grooming, RO = Rolling.) b) This resulted in human observers and machine classifiers achieving mean accuracies of 0.71 ± 0.11 (orange line) and 0.76 ± 0.069 (blue line), respectively. Statistically significant differences between these means were not noted (p > 0.05). c) Similarly, human observers and machine classifiers achieved mean ICC values of 0.93 ± 0.069 and 0.89 + 0.069, respectively.

We evaluated the influence of behavior duration on the accuracy of both human observers and machine classifiers. 37% of the total comparison dataset contained behaviors less than one second in duration, 34% were less than 0.5 seconds duration, and 14% were less than 0.1 seconds duration (Fig 6A). For both observers and classifiers, accuracy sharply increased with duration between 0.2 and 1 seconds, then plateaued at ~2.5 seconds. Human observers had 16.1 greater odds of achieving the mean overall performance benchmarks when the duration of the behavior was greater than one second (p = 0.028). Machine classifiers demonstrated 8.5 greater odds (p = 0.0059) at correctly identifying behaviors greater than one second in duration (Fig 6B).

### Influence of enrichment interventions on repetitive captive jerboa behavior

Training the classifier on the enrichment subset ended when validation accuracy exceeded the threshold value of 0.71, reaching 0.73 at the end of the final epoch. When performing inference on the comparison set, the trained classifier accuracy was 0.78 across all behaviors, despite lower accuracy in the previous section. Based on previously determined benchmarks, this suggests that the classifier achieved human-grade observation with confidence-based review and supervised learning (Table 1).

To achieve these benchmarks, the final training set consisted of 460 total minutes of manually annotated data that was then used to make inferences on ~2360 minutes of data. The supplemental data were selected to provide more training data for the behaviors with the lowest accuracy.

Repeated ethograms before the introduction of each novel enrichment intervention demonstrated individual standard deviation of 11.7 ± 2.2% across all behaviors. We then analyzed the effect of the enrichment intervention on incidence of behaviors reported by this trained classifier, measured as the difference from each individual's baseline rate. We fit a generalized linear mixed effects model ($R^2 = 0.43$) to predict behavioral incidence with enrichment type (fixed effect)

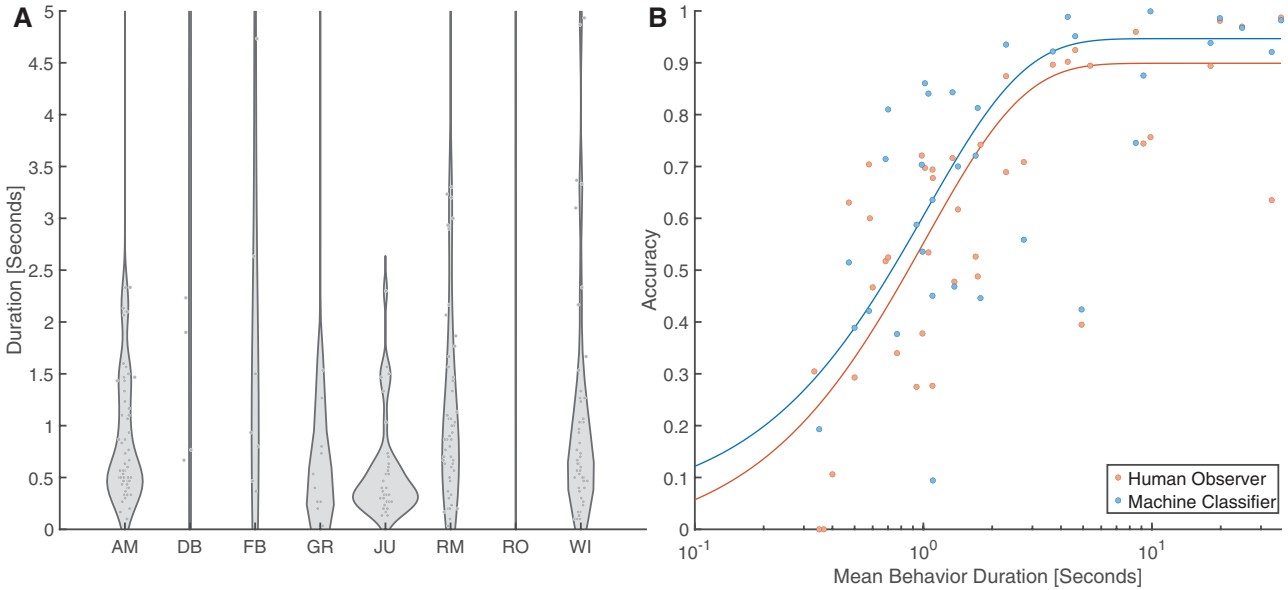

**Fig 6. The majority of jerboa behaviors are rapid, and the duration of behaviors significantly influences observer and classifier annotation accuracy.** a) Duration for each unique expression of a behavior within the comparison subset (AM = Ambulating, WI = Wall Interaction, RM = Rapid, Upright Movements, FB = Food Bowl Interactions, JU = Jumping, DB = Dust Bath Interactions, GR = Grooming, RO = Rolling.) b) Accuracy of annotating behaviors within the comparison subset increased with the mean duration for each behavior (n = 11 human observers, n = 11 machine classifiers). The data for both humans ($R^2$ = 0.58) and classifiers ($R^2$ = 0.33) were fit to exponential decay functions for visual representation of trends.

and individual (random effect). Cardboard tunnel (p = 4.6 x10⁻⁵, CI [0.53 1.5], estimate = 1) and manzanita wood stick (p = 0.033, CI [0.035 0.82], estimate = 0.43) interactions significantly increased with the presence of the respective item, as expected. Grooming demonstrated the greatest statistical interaction with enrichment items (p = 5x10⁻⁷, CI [−1.48 −0.65], estimate = −1.) Within this model, the divider was significantly correlated with upright, rapid movements and significantly inversely correlated with grooming (Fig 7).

## Discussion

This work describes use of computer vision and machine learning to classify the captive behaviors of the Lesser Egyptian Jerboa (*Jaculus jaculus*) and assesses effects of environmental enrichment. We validated use of remote recording and computer vision tools in jerboas and used them to quantify activity budgets with standard husbandry. This foundational study also evaluated effectiveness of common rodent enrichment for the first time in captive jerboas.

The use of the Raspberry Pi camera system presents a cost-effective, customizable, and easily-accessible method for recording behaviors of nocturnal animals [32]. Recordings can be programmed for remote or automatic initiation, allowing for observation of spontaneous, unprovoked behaviors within the home environment without human presence. This has been demonstrated to elicit a wider range of natural behaviors [33]. This also allows for animal assessment in an environment over which they have some autonomy, control, and spend most of their time [8–10,15,34].

We also demonstrated how classifiers can use techniques like confidence-based review to achieve more consistent accuracy among trained models. Both human observers and computer classifiers initially performed poorly at identifying grooming in the validation phase, but the final enrichment phase classifier reported a significantly higher accuracy after more grooming clips were added to the training set. The confidence estimation and scalability of machine learning models are particularly useful features for quantitative evaluation of behavior, for both clinical and experimental assessment. In this study, classifiers saved the need for manual annotation of ~40 hours of recorded data. Given the annotation speed of

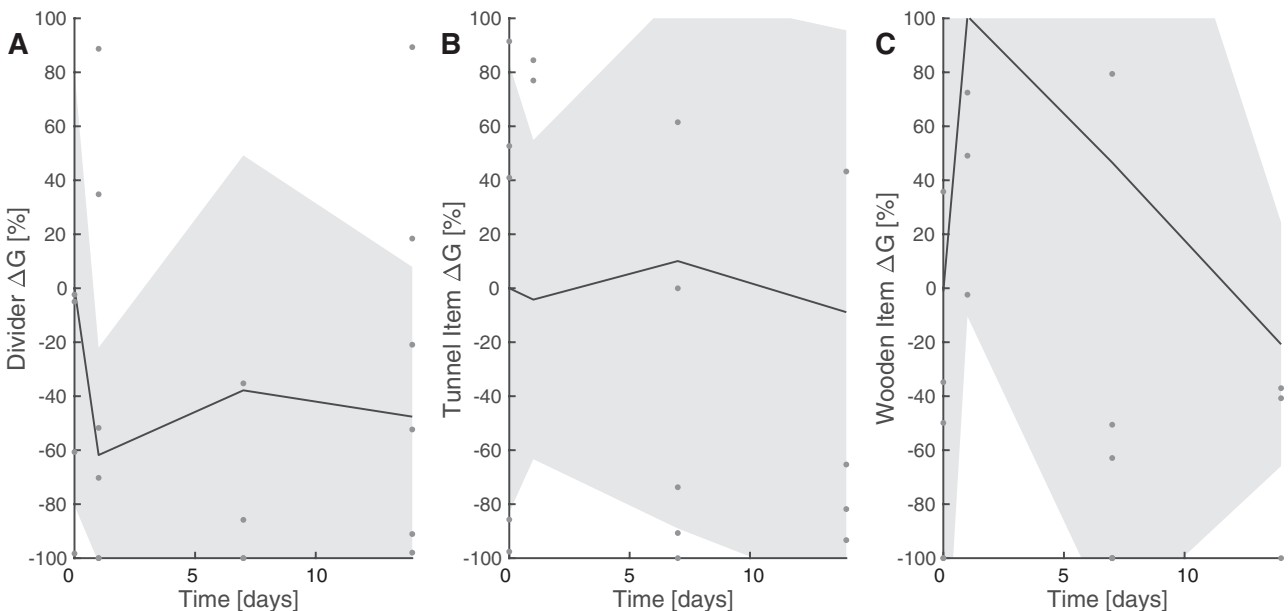

**Fig 7. The incidence of grooming is most influenced by the addition of an opaque divider, as compared to manipulanda.** Environmental manipulations included addition of (a) an opaque divider (n = 6), (b) cardboard tunnel (n = 5), or (c) manzanita stick (n = 7). ΔG is the difference in the mean number of frames spent grooming in comparison to that animal's baseline rates with standard husbandry. The solid line represents the mean of the cohort (dots), whereas shading reflects one standard deviation within the cohort. There is a dramatic decrease in grooming with the divider present, especially when compared to either baseline or the presence of other enrichment items.

human observers, this likely would have required ~200 hours of continuous effort. Jerboas demonstrated high variation both in terms of individuality, and inconsistency over time–highlighting the need for large amounts of data.

Daily activity budgets indicate that the most common behavioral states involved feed-seeking, aligning with previous descriptions of wild jerboas. Field-based descriptions of jerboa foraging similarly involve frequent movement between seed patches, rather than quickly consuming an entire seed patch [35].

Through elimination of other potential causes of alopecia and correlation with higher levels of grooming, our findings suggest jerboas exhibit psychogenic alopecia. Psychogenic alopecia, or barbering, is reported in many laboratory housed species ranging from primates to mice [25,28,31,36]. Anecdotally, alopecia in jerboas has not been associated with negative fecundity or a reduction in biomechanical performance, but may indicate boredom. We also observed that jumping had a negative correlation with alopecia. This aligns with previous observations of behavioral diversity wherein positive welfare is associated with a larger repertoire of expressed behaviors, which likely includes jumping in jerboas [10]. Future work should investigate the influence of season, light cycle, diet, and other environmental factors and extend the observation period to account for the time needed to grow new fur.

Corner jumping and digging are abnormal repetitive behaviors in captive gerbils (*Meriones unguiculatus*) [7] and deer mice (*Peromyscus leucopus*) [11]. However, in jerboas, the observed jumping was not associated with signs of distress or illness. No secondary lesions on the feet or head were noted, as might be expected with compulsive jumping. Jumping was also readily interruptible, as indicated by the short durations. Wild jerboa pups engage in jumping behavior without apparent purpose, which is likely a form of locomotor play, as seen in other active prey species (e.g., deer, *Odocoileus* spp.) [14,37–43]. While play behavior is most often associated with juvenile animals learning to interact socially, there are many types of play exhibited by adults, including locomotor or frolicking play, such as rabbit "binkying" [44]. In jerboa, jumping may also increase evolutionary fitness by maintaining hindlimb muscles necessary for rapid acceleration [12].

This, alongside both a negative trend between the amount of jumping and the presence of alopecia as well as the impact of terrarium height, suggests that providing jerboa with adequate jumping opportunities could represent a significant husbandry refinement for this species.

In other species with complex behavioral repertoires, such as nonhuman primates, access to conspecifics or objects is considered necessary for appropriate welfare. However, we noticed a decrease in rates of grooming associated with opaque dividers in our study. This, combined with conspecific aggression observed during captive breeding, suggest that jerboas may be semi-social animals [23,45]. Husbandry refinements could allow jerboas to choose when to be social, which may facilitate assessment of the preferences of individual animals for social interactions at a given time [46]. Opaque dividers are a relatively simple husbandry alteration that could improve welfare in this species. When considering manipulanda, enrichment that better mimics a jerboa's natural ecology might have been more effective, as has been shown in other species [9,37,47,48].

Examining the behavior of jerboas has provided broadly applicable insights. First, machine learning and computer vision are powerful and precise tools to examine large behavioral datasets, making them especially useful for establishing ethograms and husbandry practices for an emergent model species. Next, automated classifiers outperformed human observers for shorter duration behaviors, suggesting that computer vision can especially aid in examining animals with rapid movements and transitions between behaviors. Identification of behavioral health indices should consider ecological context and evolutionary history, along with the unique anatomy and physiology of the species. In the case of jerboas, the decrease in grooming observed due to visual isolation supports the hypothesis that they have a predominantly solitary lifestyle, as suggested by anecdotal observations of jerboas in the field [49]. Furthermore, adding standard rodent enrichment did not yield significant results for jerboas, highlighting the necessity of thoughtful consideration rather than a one-size-fits-all approach–even amongst closely related species. This is in juxtaposition to our alopecia scale, which was generalized from other species and sufficiently described all observed cases. This suggests that psychogenic alopecia should be considered in various haired animals and potentially has more conserved neuroethology [26]. Overall, our results demonstrate that a data-driven approach to establish and refine behavioral metrics of animal condition and evidence-based medicine can be boosted through the incorporation of existing computer vision toolkits.

## Supporting information

**S1 Table. Mutually exclusive ethogram for captive singly housed jerboa during the dark photoperiod utilized for observer rating.**
(DOCX)

**S2 Table. Summary of individual jerboa enrichment item introduction schedule.**
(DOCX)

**S3 Table. Description of additional exclusive ethogram terms with the addition of enrichment items.**
(DOCX)

**S4 Table. Summary of enrolled animals.**
(DOCX)

## Acknowledgments

The authors would like to thank the animals and their caretakers at the University of Michigan. We would also like to thank the Pathology Core (RRID:SCR_018823), run by the Unit for Laboratory Animal Medicine (ULAM) for services of tissue processing and histopathology.

## Author contributions

**Conceptualization:** Matthew D. Boulanger, Tara Martin, Gerry Hish.

**Data curation:** Matthew D. Boulanger, Juri A. Miyamae, Gerry Hish.

**Formal analysis:** Matthew D. Boulanger, Gerry Hish, Talia Y. Moore.

**Investigation:** Tara Martin, Gerry Hish.

**Methodology:** Matthew D. Boulanger, Juri A. Miyamae, Tara Martin, Gerry Hish, Talia Y. Moore.

**Project administration:** Gerry Hish, Talia Y. Moore.

**Resources:** Matthew D. Boulanger, Talia Y. Moore.

**Software:** Gerry Hish.

**Supervision:** Juri A. Miyamae, Tara Martin, Gerry Hish, Talia Y. Moore.

**Validation:** Matthew D. Boulanger, Gerry Hish.

**Visualization:** Matthew D. Boulanger, Gerry Hish.

**Writing – original draft:** Matthew D. Boulanger.

**Writing – review & editing:** Matthew D. Boulanger, Juri A. Miyamae, Tara Martin, Gerry Hish, Talia Y. Moore.

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
