## [Decision Letter · Decision Letter 0]

8 Sep 2025

Dear Dr. Moore,

Thank you for submitting your manuscript to PLOS ONE. After careful consideration, we feel that it has merit but does not fully meet PLOS ONE’s publication criteria as it currently stands. Therefore, we invite you to submit a revised version of the manuscript that addresses the points raised during the review process.

We have been lucky enough to receive two sets of reviews on your manuscript. Both reviewers highlight that this is a strong manuscript with considerable application to the field of behavioural science. Both reviewers note that revisions are required, but the required revisions are relatively small overall. 

We look forward to receiving your revised manuscript.

Kind regards,

James Edward Brereton, MSc

Academic Editor

PLOS ONE

4. We note you have included a table to which you do not refer in the text of your manuscript. Please ensure that you refer to Table 1 in your text; if accepted, production will need this reference to link the reader to the Table.

5. We notice that your supplementary tables are included in the manuscript file. Please remove them and upload them with the file type 'Supporting Information'. Please ensure that each Supporting Information file has a legend listed in the manuscript after the references list.

Additional Editor Comments:

Reviewer #1:

Reviewer #2:

Reviewers' comments:

Reviewer's Responses to Questions

**Comments to the Author**

1. Is the manuscript technically sound, and do the data support the conclusions?

Reviewer #1: Yes

Reviewer #2: Yes

2. Has the statistical analysis been performed appropriately and rigorously?

Reviewer #1: Yes

Reviewer #2: Yes

3. Have the authors made all data underlying the findings in their manuscript fully available?

Reviewer #1: Yes

Reviewer #2: Yes

4. Is the manuscript presented in an intelligible fashion and written in standard English?

Reviewer #1: Yes

Reviewer #2: Yes

Reviewer #1: Line 62, not everyone would know what "Old World" means

Line 83, the sentence starts with a connecting word, consider changing this

Line 132, is it still enrichment if it is standard? If it is standard then is it enriching?

Line 136, is this light cycle to mimic their habitat?

Line 272, tense changes from past to present "are"

Reviewer #2: Dear Authors,

First, I would like to thank you for the opportunity to review such an interesting piece of work. Congratulations on developing a study that is not only highly relevant for improving the management of the species but also valuable in demonstrating the suitability of using machine learning to support the collection of behavioral data.

In my view, the manuscript requires only minor adjustments to further clarify the information for readers. Below, I outline my suggestions:

Title: While the current title is interesting, I wonder whether it might be appropriate to modify it slightly. The main objective of the study appears to be the evaluation of alopecia in the animals, relating it to behavior and environmental enrichment. The comparison of methods (human vs. machine) is certainly present, but it does not seem to be the primary focus. It may therefore be worth considering a title that more directly reflects this emphasis. For example, I suggest the following:

Refining Animal Care Through Technology: Addressing Alopecia in Jaculus jaculus via Validated Computer Vision Analysis.

Line 126: What does the acronym AAALAC stand for?

Line 238: Is Equation 1 correct? It currently appears only with question marks.

Line 248: Why was the opaque divider chosen as environmental enrichment? Was it intended to block the animals’ view of conspecifics? Were the animals kept side by side with visual contact at all times, or was this arrangement only applied to the test animals?

General comment on the text: While the overall structure of the manuscript is coherent, some sentences seem more appropriate in different sections. For instance, the sentences in lines 286–288, 298–301, and 320 have more of a Discussion character, whereas the sentence in lines 289–291 appears more suited to the Methods section.

Lines 314–315: It might be clearer to rephrase as: “…and presence of grooming (statistics)….”

Table 1: I could not find where this table is first cited in the text (possibly around line 352).

Line 415: spp. should not be italicised.

Figure 4: Were there no individual differences in the expression of behaviours in these analyses? Some individuals seem to show markedly lower RM and FB values compared to others.

Figure 5: The ICC values appear to be inverted here—should they not be 0.93 ± 0.075 and 0.90 ± 0.069? In this figure, is it correct to interpret that the behaviours JU and GR showed low accuracy in recording?

**Do you want your identity to be public for this peer review?** For information about this choice, including consent withdrawal, please see our Privacy Policy

Reviewer #1: No

Reviewer #2: No

---

## [Author Response · Author response to Decision Letter 1]

22 Oct 2025

Dear Reviewers and Members of the Editing Committee,

Thank you for your time, consideration, and thoughtful feedback. Four new documents are provided in reply: this response letter, a clean document (both as a .docx and .pdf), and a document with markings to indicate changes. Within the marked-up document, text to be deleted is highlighted, red, and struck through (sample.) New text is then simply highlighted (sample). Please see responses to specific comments as follows:

Editor:

Comment: Please ensure that your manuscript meets PLOS ONE's style requirements, including those for file naming.

Response: The sizing of headers and indentation style of references have been altered to reflect the provided documents. Titles have been removed from the author list on the title page, and abbreviations have been removed from the affiliations. Table 1 has been included directly after the paragraph in which it was first cited (lines 421-424.)

Comment: When completing the data availability statement of the submission form, you indicated that you will make your data available on acceptance.

Response: The data embargo has been lifted, making the data available at the provided link.

Comment: Please include a separate caption for each figure in your manuscript.

Response: Captions have been included directly after the paragraph in which figures are first cited.

Comment: We notice that your supplementary tables are included in the manuscript file. Please remove them and upload them with the file type 'Supporting Information'. Please ensure that each Supporting Information file has a legend listed in the manuscript after the references list.

Response: The supplemental information have been removed and uploaded separately, and a legend has been included after the references.

Reviewer One:

Comment: Line 62, not everyone would know what "Old World" means

Response: The term has been removed, as it may cause confusion and this background does not seem to contribute directly to the work. Instead, the taxonomic family has been indicated on line 67 to supplement context.

Comment: Line 83, the sentence starts with a connecting word, consider changing this

Response: This sentence (now on line 89) has been edited, replacing ‘because’ with ‘so’ later in the sentence.

Comment: Line 132, is it still enrichment if it is standard? If it is standard then is it enriching?

Response: Excellent point, our use of ‘standard’ in this sentence refers to The Guide for the Care and Use of Laboratory Animals’ need for enrichment protocols. The items provided are IACUC-approved, and the current recommendation based on published guidelines1 as well as years of experience. However, the lack of available data of effective jerboa enrichment drove our study to further investigate effective enrichment for these animals. To clarify this, text has been modified on lines 138-140.

1: Jordan B, Vercammen P, Cooper KL. Husbandry and Breeding of the Lesser Egyptian Jerboa, Jaculus jaculus. Cold Spring Harb Protoc. 2011;2011: pdb.prot066712. doi:10.1101/pdb.prot066712

Comment: Line 136, is this light cycle to mimic their habitat?

Response: This light cycle is based on the only available peer-reviewed resource for jerboa husbandry.1

Comment: Line 272, tense changes from past to present "are"

Response: Thank you for catching this, the tense has been changed on line 302-303 to reflect the analysis methods versus presentation of results within the paper.

Reviewer Two:

Comment: Title: While the current title is interesting, I wonder whether it might be appropriate to modify it slightly. The main objective of the study appears to be the evaluation of alopecia in the animals, relating it to behavior and environmental enrichment. The comparison of methods (human vs. machine) is certainly present, but it does not seem to be the primary focus. It may therefore be worth considering a title that more directly reflects this emphasis. For example, I suggest the following:

Refining Animal Care Through Technology: Addressing Alopecia in Jaculus jaculus via Validated Computer Vision Analysis.

Response: Thank you for the suggestion and pointing out potential confusion, the title has been altered.

Comment: Line 126: What does the acronym AAALAC stand for?

Response: AAALAC, International is a global organization that provides voluntary accreditation to programs utilizing animals in research. The original acronym was coined in 1996 as the ‘Association for Assessment and Accreditation of Laboratory Animal Care’ International. However, in 2016 AAALAC ceased using the acronym and instead uses AAALAC International as the organization’s official, legal name.2

2: AAALAC International. History. https://www.aaalac.org/about/history/

Comment: Line 238: Is Equation 1 correct? It currently appears only with question marks.

Response: In our document, we do not see question marks in Equation 1, which suggests a potential file formatting issue. A Word document was required at submission, but we are including a clean copy of the manuscript as a PDF so you can see what the equation looks like from our perspective.

Comment: Line 248: Why was the opaque divider chosen as environmental enrichment? Was it intended to block the animals’ view of conspecifics? Were the animals kept side by side with visual contact at all times, or was this arrangement only applied to the test animals?

Response: The opaque divider blocked an animal’s view of conspecifics. Otherwise, all animals could see any immediately adjacent neighbors, as is standard in the husbandry guidelines. Animals always had at least one neighbor present, and dividers were always added on both sides of the terrarium to remove visual access. This is not entirely novel – in some species of anole lizard, opaque dividers are considered standard and a reference has been added to the body of the manuscript. In general, context for all enrichment items provided has been added on lines 277-279.

Comment: General comment on the text: While the overall structure of the manuscript is coherent, some sentences seem more appropriate in different sections. For instance, the sentences in lines 286–288, 298–301, and 320 have more of a Discussion character, whereas the sentence in lines 289–291 appears more suited to the Methods section.

Response: Thank you for catching that discrepancy. What was on lines 289-291 has been moved to Methods from Results for clarification to lines 168-170. However, we feel that the content outlined previously on lines 286–288, 298–301, and 320 is best suited within the Results section. While we compare the results to the hypotheses here, the significance of refuting hypotheses is within the Discussion section.

Comment: Lines 314–315: It might be clearer to rephrase as: “…and presence of grooming (statistics)….”

Response: The text has been rephrased as suggested on line 356-357.

Comment: Table 1: I could not find where this table is first cited in the text (possibly around line 352).

Response: Table 1, the result of the referenced training set detailed on lines 426-429, is referenced on line 420 has has been added to lines 421-424.

Comment: Line 415: spp. should not be italicized.

Response: Thank you for finding this, it has been corrected on line 495.

Comment: Figure 4: Were there no individual differences in the expression of behaviours in these analyses? Some individuals seem to show markedly lower RM and FB values compared to others.

Response: Generally speaking, there was a high degree of both inter- and intra-individual variance of behaviors across all behaviors. Figure 4 has about 11% + 4% standard deviation of behavioral incidence throughout the cohort (inter-individual). We grouped by known qualities – cage height, location on the rack, sex, and age, when performing analysis. While there is a large individual variance, different means were not significant by grouping factors beyond what is mentioned (terrarium height.) However, we feel the high individual variance speaks to the need for more advanced tools for behavioral assessment. We have added text to the manuscript to more clearly discern inter- from intra-individual variation.

Comment: Figure 5: The ICC values appear to be inverted here—should they not be 0.93 ± 0.075 and 0.90 ± 0.069? In this figure, is it correct to interpret that the behaviours JU and GR showed low accuracy in recording?

Response: Thank you for catching this clerical error – machine classifier ICC was reported with reduced significant figures. The ICC is as-reported in the manuscript as 0.89 + 0.069 (not 0.9 + 0.075). In this figure, both jumping (JU) and grooming (GR) demonstrated low accuracy when annotated by both humans and classifiers. While this could appear troubling, as both became of primary focus, confidence-based review captured more information within the extended training set, and the final classifier achieved much higher accuracies for both behaviors – 0.89 (JU) and 0.75 (GR) specifically (Table 1). We modified the text in the manuscript to make this clearer.

Thank you again for your time and insightful feedback. Please do not hesitate to contact us with additional questions or comments.

Sincerely,

Primary Author: Dr. Matthew Boulanger, VMD

Unit for Laboratory Animal Medicine

Postdoctoral Research Fellow

mdboulan@umich.edu

Co-PI: Dr. Gerry Hish, DVM, DACLAM

Assistant Professor, Unit for Laboratory Animal Medicine (ULAM)

Program Director, ULAM Postdoctoral Clinical Training Program

gerryh@med.umich.edu

PI and Corresponding Author: Dr. Talia Moore, PhD

Assistant Professor, Robotics, Mechanical Engineering

Affiliate, Ecology and Evolutionary Biology, Museum of Zoology

University of Michigan, Ann Arbor

taliaym@umich.edu

---

## [Editor Report · Decision Letter 1]

23 Oct 2025

Refining animal care through technology: addressing alopecia in Jaculus jaculus with validated computer vision analysis

PONE-D-25-40567R1

Dear Dr. Moore,

We’re pleased to inform you that your manuscript has been judged scientifically suitable for publication and will be formally accepted for publication once it meets all outstanding technical requirements.

Kind regards,

James Edward Brereton, MSc

Academic Editor

PLOS ONE
---

## [Editor Report · Acceptance letter]

PONE-D-25-40567R1

PLOS ONE

Dear Dr. Moore,

I'm pleased to inform you that your manuscript has been deemed suitable for publication in PLOS ONE. Congratulations! Your manuscript is now being handed over to our production team.

Kind regards,

on behalf of

Mr. James Edward Brereton

Academic Editor

PLOS ONE